# PeerJ

# Major histocompatibility complex class I evolution in songbirds: universal primers, rapid evolution and base compositional shifts in exon 3

Miguel Alcaide[1], Mark Liu and Scott V. Edwards

Department of Organismic and Evolutionary Biology and Museum of Comparative Zoology, Harvard University, Cambridge, MA, USA
[1] Present address: Department of Zoology, University of British Columbia, Vancouver, Canada

## ABSTRACT

Genes of the Major Histocompatibility Complex (MHC) have become an important marker for the investigation of adaptive genetic variation in vertebrates because of their critical role in pathogen resistance. However, despite significant advances in the last few years the characterization of MHC variation in non-model species still remains a challenging task due to the redundancy and high variation of this gene complex. Here we report the utility of a single pair of primers for the cross-amplification of the third exon of MHC class I genes, which encodes the more polymorphic half of the peptide-binding region (PBR), in oscine passerines (songbirds; Aves: Passeriformes), a group especially challenging for MHC characterization due to the presence of large and complex MHC multigene families. In our survey, although the primers failed to amplify exon 3 from two suboscine passerine birds, they amplified exon 3 of multiple MHC class I genes in all 16 species of oscine songbirds tested, yielding a total of 120 sequences. The 16 songbird species belong to 14 different families, primarily within the Passerida, but also in the Corvida. Using a conservative approach based on the analysis of cloned amplicons ($n = 16$) from each species, we found between 3 and 10 MHC sequences per individual. Each allele repertoire was highly divergent, with the overall number of polymorphic sites per species ranging from 33 to 108 (out of 264 sites) and the average number of nucleotide differences between alleles ranging from 14.67 to 43.67. Our survey in songbirds allowed us to compare macroevolutionary dynamics of exon 3 between songbirds and non-passerine birds. We found compelling evidence of positive selection acting specifically upon peptide-binding codons across birds, and we estimate the strength of diversifying selection in songbirds to be about twice that in non-passerines. Analysis using comparative methods suggest weaker evidence for a higher GC content in the 3rd codon position of exon 3 in non-passerine birds, a pattern that contrasts with among-clade GC patterns found in other avian studies and may suggests different mutational mechanisms. Our primers represent a useful tool for the characterization of functional and evolutionarily relevant MHC variation across the hyperdiverse songbirds.

Corresponding author
Scott V. Edwards,
sedwards@fas.harvard.edu

## INTRODUCTION

Genes of the Major Histocompatibility Complex (MHC) have become one of the most sought-after molecular markers for the investigation of adaptive genetic variation in vertebrates (e.g., *Eizaguirre et al., 2012*; *Kamath & Getz, 2011*; *Kubinak et al., 2012*; *Radwan et al., 2012*). MHC genes are known to play a critical role during the development of immunity against invading and potentially harmful pathogens. The cell-surface proteins encoded by MHC genes bind and present short peptides (antigens) derived from the processing of pathogens to lymphocyte T-cells, which triggers the adaptive branch of the immune system (*Iwasaki & Medzhitov, 2010*). MHC genes are also thought to play important roles in avian mate choice, although the conclusions of various studies have been mixed, in part because of the complexity of this redundant multigene family (e.g., *Strandh et al., 2012*; *Bollmer et al., 2012*; *Juola & Dearborn, 2011*; *Knafler et al., 2012*; *Ekblom et al., 2004*; *Westerdahl, 2004*).

MHC molecules have been traditionally classified into two major groups on the basis of the origin of the antigens presented, although some degree of cross-presentation between MHC classes is now currently assumed (e.g., *Iwasaki & Medzhitov, 2010*). Typically, MHC class I molecules are monomeric proteins known to mostly present antigens derived from intracellular pathogens (such as viruses) while MHC class II molecules are dimeric proteins deploying antigens from extracellular pathogens such as bacteria (reviewed by *Sommer, 2005*). Given the extraordinary richness and diversity of continuously evolving pathogens in the environment, it is not surprising that the MHC harbors the most polymorphic genes described thus far, with some loci, such as the human HLA-B locus, possessing more than 2,000 alleles (*de Bakker & Raychaudhuri, 2012*). The maintenance of such astonishing diversity is believed to be driven primarily by two main types of balancing selection: heterozygote advantage, by which heterozygous individuals respond better to infection than homozygous individuals, and frequency-dependent selection, by which rare, low-frequency alleles might provide a selective advantage once pathogens have found a way to elude the most common immune defense alleles in the population. The evolutionary implications of MHC variation during the pathogen-host arms race have been widely investigated across a large variety of taxa. As a result, MHC genes have been of great interest in evolutionary biology and conservation genetics, as the capability of species and populations to counter and adapt to novel pathogen menaces is believed to be tightly linked to their degree of MHC variability (see *Piertney & Oliver, 2006*; *Sommer, 2005*; *Spurgin et al., 2011*), but see also (*Gangoso et al., 2012*; *Radwan, Biedrzycka & Babik, 2010*; *Westerdahl et al., 2012*).

Despite their great interest and potential for ecological immunology, the isolation and characterization of MHC genes in non-model species still remains a challenging and

**Figure 1 Schematic representation of part of an MHC class I gene.** Arrows indicate the location of the primers used in this study. Both the coding sequences of exon 2 and exon 3 comprise the antigen-binding region of MHC class I molecules. Exons are represented by boxes and the lines connecting boxes represent introns.

time-consuming task. Until recently, the description of MHC genes in birds, particularly for class I genes, was mostly restricted to galliform species and a very few species of songbirds (reviewed by *Hess & Edwards, 2002*; *Westerdahl, 2007*). The last few years have nevertheless witnessed great progress regarding the isolation and characterization of MHC genes in non-model avian species, particularly across the avian MHC class II B multigene family (e.g., *Alcaide, Edwards & Negro, 2007*; *Burri et al., 2008*; *Canal et al., 2010*; *Ekblom, Grahn & Hoglund, 2003*; *Li, Zhou & Chen, 2011*; *Silva & Edwards, 2009*; *Strandh et al., 2011*). Additionally, significant advances in genotyping protocols for complex multigene families like the MHC (reviewed by *Babik, 2010*) have facilitated the MHC studies in general, particularly in songbirds displaying large number of MHC gene paralogs (e.g., *Bollmer et al., 2010*; *Sepil et al., 2012*; *Zagalska-Neubauer et al., 2010*). Studies addressing MHC class I variability in birds are, however, less numerous or phylogenetically diverse than those for MHC class II B genes (see recent examples in *Cloutier, Mills & Baker (2011)*, *Promerova, Albrecht & Bryja (2009)*, *Sepil et al. (2012)* and *Westerdahl (2004))* in part almost certainly due to the lack of suitable primers for the cross-amplification of candidate loci across species (see however *Alcaide et al., 2009*). Here, we describe the utility of a single pair of primers for the cross-amplification of MHC class I loci across a large avian order (Aves: Passeriformes), which includes more than half of known avian species, in an effort to facilitate the study of MHC variation in non-model avian species. From the two exons of MHC class I genes (exon 2 and 3) that comprise the peptide-binding region (PBR), we focused on exon 3, in part because it has generally been found to be the more polymorphic of the two (e.g., *Cloutier, Mills & Baker, 2011*). Putting our data together with published class I sequences from non-passerine birds, we analyzed the macroevolution of both rates of adaptive evolution and base compositional shifts as windows into the selective and mutational pressures experienced by songbird class I genes.

## MATERIALS AND METHODS

We used the QIAGEN–DNeasy Blood & Tissue Kit (Qiagen, CA, USA) to obtain genomic DNA from blood samples collected in the field and tissue samples from the Museum of Comparative Zoology at Harvard University (Cambridge, MA, USA; Animal protocol number AEP 24-06) following the manufacturer's protocol. The list of songbird species investigated is shown in Table 1. We initially amplified the entire coding sequence of exon 3 of MHC class I genes and a small part of the flanking intronic regions (see Fig. 1) in two house finches (*Haemorhous mexicanus*) and two Eastern Bluebirds

**Table 1 Amplification success and genetic variability within each of the 16 oscine songbird and two suboscine species here investigated.** The table shows the putative number of functional alleles per species (Na; the number of putative pseudogene sequences, if any, is given in parentheses), the overall number of polymorphic sites per allele repertoire (S), average nucleotide diversity ($\pi$) and average number of nucleotide differences ($k$) among the sequences isolated from the same species. This table also shows the ratio ($\omega$) between non-synonymous ($d_n$) and synonymous ($d_s$) substitution rates for those codons presumably comprising (PBR) and non-comprising (non-PBR) the peptide binding region of the MHC class I molecule (see text for details). Sample sizes are 1 for all species except for the Eastern Bluebird and house finch, for which $n = 2$. The accession numbers for the specimens from the Museum of Comparative Zoology (MCZ) ornithology collection from which DNA was isolated are given. n/a, not accessioned. –, no amplification.

| Latin name | Common name | Family | MCZ no. | Na | S | $\pi$ | K | $\omega = dn/ds$ PBR | $\omega = d_n/d_s$ non-PBR |
|---|---|---|---|---|---|---|---|---|---|
| *Passer domesticus* | House sparrow | Passeridae | 337599 | 8 | 56 | 0.1 | 25.86 | 3.69 | 0.53 |
| *Cardinalis cardinalis* | Northern cardinal | Cardinalidae | 337661 | 7 | 68 | 0.124 | 32.9 | 3.08 | 0.45 |
| *Thraupis episcopus* | Blue-grey tanager | Thraupidae | 337677 | 8 (3) | 103 | 0.16 | 41.82 | 4.16 | 0.74 |
| *Bombycilla cedrorum* | Cedar waxwing | Bombycillidae | 337636 | 9 | 52 | 0.086 | 22.7 | 1.02 | 0.31 |
| *Agelaius phoeniceus* | Red-winged blackbird | Icteridae | 337415 | 8 | 63 | 0.105 | 27.5 | 6.83 | 0.79 |
| *Sturnus vulgaris* | European starling | Sturnidae | 337556 | 9 | 108 | 0.162 | 42.44 | 2.46 | 0.58 |
| *Thryothorus thoracicus* | Stripe-breasted wren | Troglodytidae | 337696 | 6 (3) | 56 | 0.107 | 28.47 | 7.12 | 0.72 |
| *Turdus migratorius* | American robin | Turdidae | 337189 | 3 (3) | 56 | 0.145 | 33.8 | 1.89 | 0.69 |
| *Geothlypis trichas* | Common yellowthroat | Parulidae | 337642 | 5 | 63 | 0.133 | 34.8 | 3.72 | 0.47 |
| *Dumetella carolinensis* | Gray catbird | Mimidae | 337601 | 5 | 66 | 0.129 | 33.7 | 1.56 | 0.65 |
| *Passerina cyanea* | Indigo bunting | Cardinalidae | 337535 | 10 | 88 | 0.137 | 35.78 | 2.79 | 0.45 |
| *Polioptila plumbea* | Tropical Gnatcatcher | Polioptilidae | 337547 | 6 | 33 | 0.056 | 14.67 | 0.21/0.00 | 1.07 |
| *Vireo olivaceus* | Red-eyed vireo | Vireonideae | 337166 | 6 | 92 | 0.165 | 43.67 | 3.53 | 0.76 |
| *Sitta canadensis* | Red-breasted nuthatch | Sittidae | 337181 | 6 | 34 | 0.057 | 15 | 0.13/0.00 | 0.63 |
| *Sialia sialis* | Eastern Bluebird | Turdidae | n/a | 13 | 80 | 0.129 | 33.8 | 0.91 | 0.40 |
| *Haemorhous mexicanus* | House finch | Fringillidae | n/a | 11 | 85 | 0.149 | 38.8 | 5.03 | 0.65 |
| *Sayornis phoebe* | Eastern Phoebe | Tyrannidae | 337162 | – | – | – | – | – | – |
| *Manacus candei* | White-Collared Manakin | Pipridae | 348105 | – | – | – | – | – | – |

(*Sialia sialis*) using primers HN34 (5′-CCATGGGTCTCTGTGGGTA-3′) and HN45 (5′-CCATGGAATTCCCACAGGAA-3′) from *Westerdahl et al. (2004)*. Although these primers were originally designed for the isolation of MHC class I loci in great reed warblers (*Acrocephalus arundinaceus*) they have proven successful in the isolation of MHC class I sequences in other passerine species (e.g., *Promerova, Albrecht & Bryja, 2009*; *Sepil et al., 2012*). PCR amplification was carried out using a PTC-100 Programmable Thermal Controller (MJ research, MA, USA) in a final volume of 25 µl containing 1 unit of EconoTaq DNA polymerase (Lucigen Corporation, Middleton, WI, USA), 1× PCR buffer (Lucigen), 1 mM MgCl$_2$, 10 pmoles of each primer, 0.2 mM of each dNTP, 10 µg of BSA, 5% DMSO, and approximately 10–30 ng of DNA. The cycling protocol consisted of an initial denaturation step of 95° during 4 min, followed by 35 cycles of 95°C for 45 s, 55°C for 45 s and 72°C for 45 s plus a final extension step of 72°C during 4 min. After visualization of PCR products in 1% agarose gels stained with SYBR safe (Invitrogen, CA, USA) we cloned fragments using the StrataClone PCR cloning kit (Agilent Technologies, Inc., CA, USA) and inserts of the expected size (around 350 bp) were re-amplified using M13 primers, purified with ExoSAP-IT reagent (Affymetrix, CA,

USA) and sequenced with M13 primers and BigDye 3.1 reagents (Applied Biosystems, CA, USA) supplemented with BDX64 buffer (MCLAB, CA, USA) according to the manufacturer's protocols. Sixteen positive clones from each of the two birds were selected for sequencing. Fluorescently labeled fragments were resolved into an Applied Biosystems 3730 xl DNA Analyzer (Applied Biosystems, CA, USA). The sequences obtained from house finches and Eastern Bluebirds were aligned in BioEdit ver 7.0 (*Hall, 1999*) and Geneious R6 (*Drummond et al., 2011*). New degenerate primers sitting on the boundaries of exon 3 were designed following the alignment of house finch and Bluebirds MHC class I sequences. These primers were MhcPasCI-FW 5′-CSCSCAGGTCTSCACAC-3′ and MhcPASCI-RV 5′-CWCARKAATTCTGYTCHCACC-3′ (Fig. 1). Primer MhcPasCI-FW is similar in its sequence to the primers HN36 and HN38 designed by *Westerdahl et al. (2004)* but our primer is more degenerated and 3 nucleotides shorter in the 3′ end (the last nine nucleotides of the primer sit into the coding sequence of exon 3). The utility of this primer pair was tested in each of the 16 songbird species from 14 different taxonomic families, as well as two suboscine passerine species, a manakin (Pipiridae) and a New World Flycatcher (Tyrranidae). The PCR protocol used was the same as that described above for the primer pair HN34/HN45. Likewise, PCR products were cloned and sixteen positive bands per species or individual sequenced with M13 primers as described above. Sequences were again edited and aligned in BioEdit and Geneious R6 (*Drummond et al., 2011*). We then searched for different, putative alleles within each individual. Given the low number of clones screened we only considered as different, presumably functional alleles, those DNA sequences differing in at least three nucleotide positions and not showing stop codons or disrupted reading frames. These criteria will minimize the impact of PCR and sequencing artifacts in our allele repertoire but may also underestimate the number of alleles per individual (e.g., when true alleles differed in just one nucleotide positions but they are not found in more than one clone each). All sequences fitting these criteria have been deposited into the GenBank public domain (see results). Putative pseudogenes and those sequences suspected to be mosaic or chimeric sequences were discarded. In a study focused on population genetics, the genotyping of MHC loci across large complex multigene families requires more stringent criteria and protocols to define true alleles than those described in this study (e.g., *Lenz & Becker, 2008*; *Sepil et al., 2012*). Here our purpose was to produce a first glimpse of the utility of our primers and broad macroevolutionary patterns, but future studies using our primers should more thoroughly analyze a MHC variability in particular species with more stringent quality checks.

The MHC class I sequences here isolated plus additional exon 3 sequences from non-passerine species downloaded from the public domain (see File S1) were aligned using the Muscle algorithm (*Edgar, 2004*) as implemented in TranslatorX, a codon-based alignment algorithm (*Abascal, Zardoya & Telford, 2010*), using default options. We built a phylogenetic tree using the Neighbour-joining method (*Saitou & Nei, 1987*) as implemented in Geneious R6 (*Drummond et al., 2011*) using an optimal substitution model (Tamura-Nei model + gamma = 0.78 in this case) from ModelTest v. 3.5 (*Posada & Crandall, 1998*). An MHC class I exon 3 sequence of the Balsas armed lizard (*Ctenosaura*

*clarki*l; GenBank Acc. No. EU839667) was used as outgroup. Branch support was evaluated by 1,000 bootstrap replicates. We verified that the major branches in this tree, including the strongly supported branches leading to the songbird sequences and to several of the major clades of sequences were present when using the maximum likelihood method as implemented in Phyml v. 3.0, using SPR tree searching and an HKY model of substitution with estimated variation in rates among sites (*Guindon et al., 2010*). The phylogenetic relationships among songbird MHC class I sequences were also visualized through a neigbor-net network built in Splitstree 4.0 (*Huson & Bryant, 2006*) according to Kimura-2-parameter distances. We tested for a significant clustering of MHC sequences by species using a permutation test in MacClade in which we compared the observed number of parsimony changes in a character coded as species designation on the best tree with the distribution observed on 1000 random trees. Non-synonymous ($d_n$) and synonymous substitution rates ($d_s$) were calculated in MEGA ver 5.0 (*Tamura et al., 2011*) according to the modified Nei-Gojorobi method with Jukes-Cantor correction and 1,000 bootstrap replicates for variance estimation. Two analyses were carried out, one including only putative PBR codons and another including the remaining codons. Codons were labeled as PBR or non-PBR in accordance with previously documented patterns of positive selection across the avian MHC class I (see *Balakrishnan et al., 2010*; *Alcaide et al., 2009*), which also suggested large overlapping between the PBR of the human MHC class I molecule (*Björkman et al., 1987*; *Saper, Bjorkman & Wiley, 1991*) and homologous sites in birds. The exon 3 codons classified as PBR-codons were 5, 7, 8, 9, 23, 25, 38, 60, 61, 62, 65, 66, 68 and 73.

We carried out an independent analysis of codon substitution using the codeml software in the PAML package v.4.4 (*Yang, 2007*). Specifically we used the branch-site and clade tests (models A, C and D, see *Bielawski & Yang, 2004*; *Yang, Wong & Nielsen, 2005*), focusing on the branch leading to songbirds (model A), or the entire clade of songbirds (models C and D). These models differ in their structure as well as in the assumptions about the values that specific parameters can take. For example, in model A, the 'background' branch cannot have any sites with values of $\omega$ greater than 1, and the test focuses on a specific branch, in our case the single branch leading to the passerine sequences. By contrast, in both models C and D, one class of sites ($\omega_0$) is constrained to fall below 1. But other classes of sites ($\omega_1$) are either constrained to equal 1 (model C) or allowed to take on any positive value (model D). Models C and D both test average rates across clades, rather than specific branches, but they differ in the statistical models assumed, with model C employing the more accurate Bayes-Empirical-Bayes (BEB) method, rather than the naïve empirical bayes (NEB) approach (*Bielawski & Yang, 2004*; *Yang, Wong & Nielsen, 2005*). However all of these methods allowed us to assess those PBR codons experiencing adaptive evolution without identifying them a priori.

We also noticed pronounced variation in GC content of the avian MHC class I sequences across various groups, particularly in the 3rd position of codons. Given that base compositional variation is a consequence of a variety of molecular forces, such as recombination, gene conversion and selection, this base compositional variation could be

important for understanding the evolutionary forces operating on avian MHC class I genes (*Duret & Galtier, 2009*). To quantify and better understand the dynamics of GC-content in avian MHC class I genes, we analyzed the phylogenetic signal in GC content using a comparative method (see *Thomas, Meiri & Phillimore, 2009*) that is well suited to testing for the significance of means and variances of continuous characters among clades under a Brownian motion model. Base compositional evolution in DNA sequences has often been challenging to analyze because standard phylogenetics packages that allow this calculation often perform standard, non-phylogenetic tests of significance, which are inappropriate (e.g., PAUP*; *Swofford, 2002*). The likelihood method of *Thomas, Meiri & Phillimore (2009)* (see also *Thomas, Freckleton & Szekely, 2006*) is well suited for the analysis of diversification of continuous phenotypic or genotypic traits. This approach is able to distinguish between two possible causes of differences in means between groups: differences in rate of trait evolution (as revealed by the Brownian variance of the trait) between groups without there being differences in means, and true differences in means between groups, with or without differences in rates of evolution. For these calculations we removed the outgroup and used an ultrametricized tree using the penalized likelihood method of *Sanderson (2002)*, using a value of 2 for the smoothing parameter λ. The Brownian motion analysis was implemented in the 'motmot' package in R, version 1.0.1 (*Thomas & Freckleton, 2012*).

## RESULTS

*Versatile primers for songbirds*: The primer pair MHCPasCI-Fw and MHCPasCI-Rv successfully amplified multiple MHC class I sequences in all 16 songbird species tested in the present study (Table 1, GenBank Acc. Nos. KC585518-KC585637, see also File S2). The number of putatively functional MHC alleles ranged from 3 to 10 per individual. This is surely an underestimate given the low number of clones screened per individual and the possibility of confounding true alleles differing by less than 3 substitutions with PCR or sequencing artifacts. However, genetic diversity within the allele repertoire isolated from the same individual or species was high. The number of variable sites ranged from 33 to 108 (out of 264) and the average number of nucleotide differences between putative alleles or loci within species ranged from 14.67 in the tropical gnatcatcher to 43.67 in the European Starling (Table 1). Non-synonymous substitution rates commonly exceeded synonymous substitution rates at those codons presumably comprising the peptide binding, while the contrary was the general pattern outside PBR codons (Table 1). This finding is consistent with our having amplified functional MHC genes subjected to positive, diversifying selection. The co-amplification of putative pseudogenes, on the other hand, seemed to be common in the following species: Blue-grey Tanager, Stripe-breasted Wren and American Robin, while in the remaining species this phenomenon appeared to be rare (Table 1). An alignment of the selected and presumably functional MHC class I sequences is included as Supplemental Information (File S1) and on Dryad and Treebase. Four out of the five MHC class I alleles that we isolated in the Common Yellowthroat (an individual captured in Massachusetts, U.S.) are identical to some of the alleles previously isolated in the same species (from a Wisconsin population, U.S.) by *Bollmer et al. (2012)*

through a pyrosequencing approach (GenBank Acc. Nos. AFP1784, AFP17830, AFP17847 and AFP17865, see Files S1 and S2). In addition, four out of the eight alleles that we isolated in the house sparrow differed in no more than three amino acid positions from other alleles previously isolated in this species by *Bonneaud et al. (2004)* and *Loiseau et al. (2008)* (GenBank Acc. Nos. AAQ22383, ABO15711). These similarities lend confidence to our results and suggest we have in some cases amplified the same or very similar loci. The absence of any identical alleles in house sparrows is likely due to the different geographic origin of the house sparrow individual here investigated (Massachussets, U.S.) versus the house sparrows individuals genotyped in the studies by *Bonneaud et al. (2004)* and *Loiseau et al. (2008)* (Europe, see for instance *Alcaide et al., 2008* for marked genetic structuring at the MHC). After repeated attempts at amplification, we nonetheless found out that our primers do not amplify a homologous MHC class I fragment from the two suboscine species that we investigated, a manakin (Pipiridae) and a New World Flycatcher (Tyrranidae) (Table 1).

*Phylogenetic relationships*: The phylogenetic relationships among the MHC class I sequences isolated in this study plus additional exon 3 sequences isolated in other avian species are depicted in Fig. 2. The resulting tree defined two main clades, one including all songbirds and the other encompassing the rest of bird species, including chicken, Anseriformes, diurnal birds of prey, petrels, seagulls, falcons and kestrels. We also found that there was strong support for clustering of sequences for various higher clades, such as Anseriformes (Mallard and goose in our data set), Falcons/Kestrels, diurnal birds of prey + gull + petrel, and chicken. The lack of complete clustering according to species for many of the 17 nonpasserine or 16 passerine species is not surprising given the expected impact of trans-species polymorphisms across the MHC in general (e.g., *van Oosterhout, 2009*) and the complex nature of the songbird MHC class I multigene family in terms of paralogs. Still, using the permutation test, we found that the observed number of parsimony transitions of a character ($x$) labeled as species designation in both songbirds ($x = 33$) and non-passerines ($x = 18$) was significantly lower than that observed among 1000 random trees (passerines, range $x = 99$–$115$ steps; non-passerines, range $x = 62$–$76$; both groups, $P < 0.001$), suggesting significant clustering by species. For example, the 15 sequences from the two tit species from the database all clustered together (although not by species), as did the 6 gnatcatcher, 6 vireo and 9 waxwing sequences (see Fig. S1 for a more detailed tree, see also Fig. 3). On the other hand, the neighbor-net depicted in Fig. 3 shows a complex network with multiple reticulate events both within and among species, particularly in a cluster of blackbird, tanager and yellowthroat sequences. Evidence for recombination using the SplitsTree Phi test was not significant ($p = 0.09$) across the entire songbird MHC class I data set, although this test is ideally applied to a large set of MHC sequences isolated from the same or closely related species. Qualitatively, our results suggest a role for recombination during the evolution of the MHC class I multigene family in songbirds.

*Greater selection intensity in the PBR of songbird MHC class I*: We analyzed patterns of nucleotide substitution using MEGA and PAML. When estimating $\omega$ using the modified
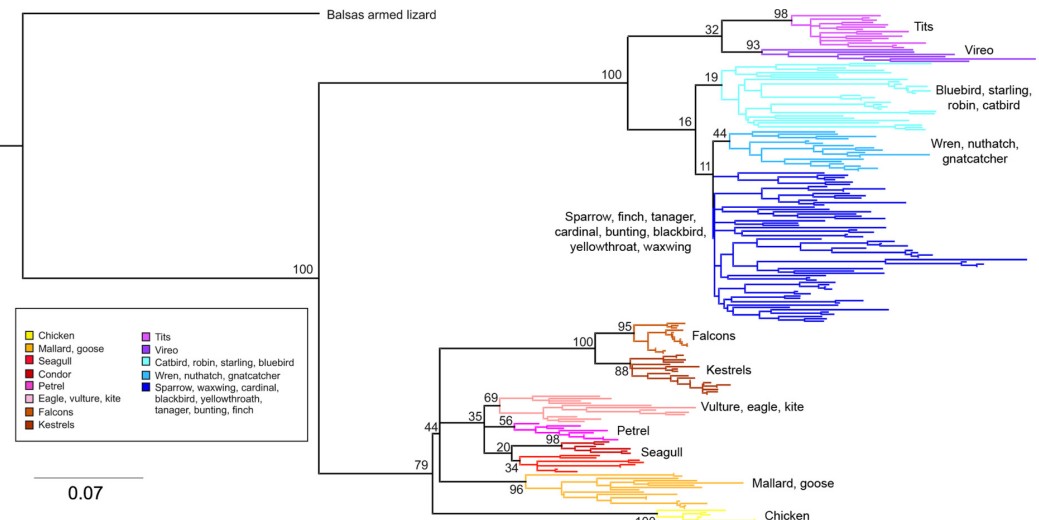

**Figure 2** Neighbor-joining tree of the passerine MHC class I sequences (exon 3) here isolated plus additional exon 3 sequences isolated in other avian species. Bootstrap support for the main branches of the tree are indicated. A more detailed depiction of this tree is provided in Fig. S1.

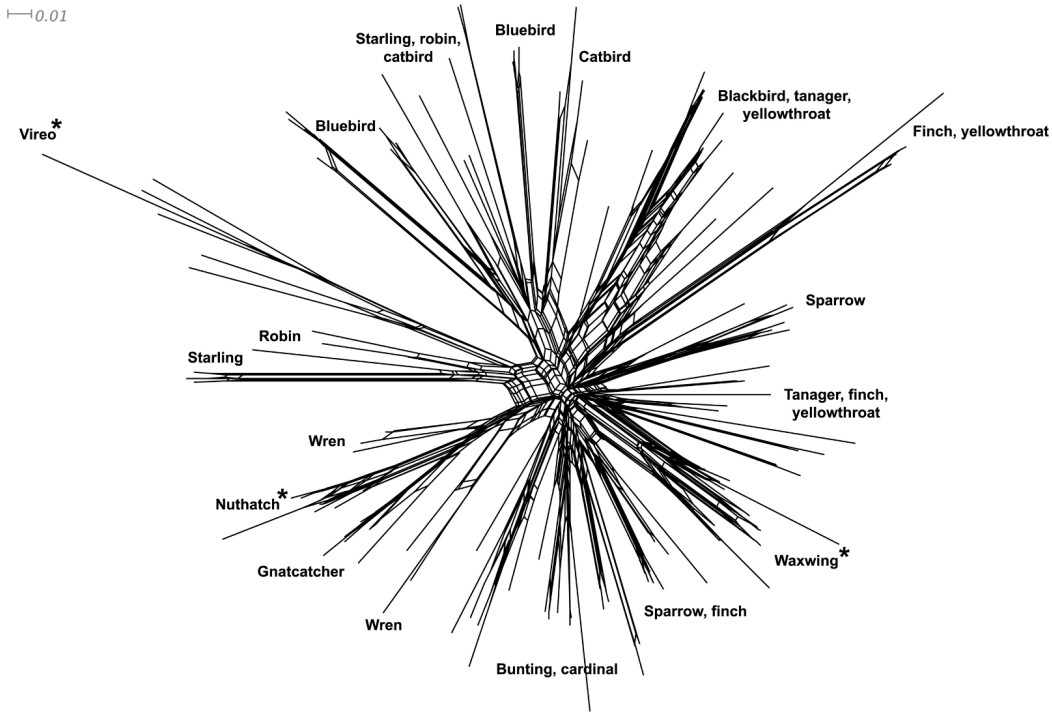

**Figure 3** Neighbor-net network of the MHC class I sequences (exon 3, $N = 120$) isolated from the 16 songbird species investigated in this study. Only the main clusters of sequences are labeled for simplicity. Those species whose sequences fall into a single cluster are indicated by asterisks.

**Table 2 Estimates of non-synonymous ($d_n$) and synonymous ($d_s$) substitution rates and their ratio for codons chosen a priori to comprise (PBR) and not comprise (non-PBR) the peptide binding region of the MHC class I molecule (MEGA) and for codon classes estimated from the data (PAML).** For each comparison estimates for songbirds (oscine passerines) and non-passerines are provided (see also Fig. 2). For the MEGA estimates, standard errors based on 1,000 bootstrap replicates are given. For the PAML estimates, the results of two models (C and D) are given. In models C and D, $\omega_0$ is constrained to fall below 1, whereas in model C, $\omega_1$ must equal 1.

| | Site class | Parameter[a] | Proportion of sites | Clade | |
| --- | --- | --- | --- | --- | --- |
| | | | | Songbirds | Non-Passeriformes |
| **MEGA** | | | | | |
| | PBR | $d_n$ | 0.791 | $0.620 \pm 0.157$ | $0.474 \pm 0.129$ |
| | | $d_s$ | 0.209 | $0.163 \pm 0.074$ | $0.246 \pm 0.111$ |
| | | $\omega = d_n/d_s$ | | 3.80 | 1.92 |
| | non-PBR | $d_n$ | 0.769 | $0.144 \pm 0.017$ | $0.135 \pm 0.022$ |
| | | $d_s$ | 0.231 | $0.255 \pm 0.041$ | $0.284 \pm 0.047$ |
| | | $\omega = d_n/d_s$ | | 0.56 | 0.48 |
| | All codons | $d_n$ | 0.772 | $0.176 \pm 0.025$ | $0.160 \pm 0.024$ |
| | | $d_s$ | 0.228 | $0.245 \pm 0.037$ | $0.278 \pm 0.044$ |
| | | $\omega = d_n/d_s$ | | 0.72 | 0.58 |
| **PAML** | | | | | |
| Model A | 0 | $0 < \omega_0 < 1$ | 0.546 | 0.230 | 0.230 |
| | 1 | $\omega_1 = 1$ | 0.301 | 1.000 | 1.000 |
| | 2a | $\omega_2 \geq 1, 0 < \omega_0 < 1$ | 0.991 | 2.479 | 0.230 |
| | 2b | $\omega_2 \geq 1, \omega_1 = 1$ | 0.055 | 2.479 | 1.000 |
| Model C | 0 | $0 < \omega_0 < 1$ | 0.380 | 0.111 | 0.111 |
| | 1 | $\omega_1 = 1$ | 0.120 | 1.000 | 1.000 |
| | 2 | $\omega_2$ | 0.500 | 0.476 | 0.421 |
| Model D | 0 | $0 < \omega_0 < 1$ | 0.381 | 0.141 | 0.141 |
| | 1 | $\omega_1$ | 0.098 | 2.388 | 2.388 |
| | 2 | $\omega_2$ | 0.521 | 2.388 | 0.633 |

**Notes.**

[a] For the PAML models, when two parameters are listed in the column, the first parameter in the cell refers to the songbird branch (model A) or clade (model C or D), and the second refers to the non-passerine branch or clade.

*Nei & Gojobori (1986)* method in MEGA, we found evidence for rates of diversifying selection acting on PBR codons in songbirds about twice as great as those found in non-passerine birds (Table 2). For non-PBR codons, the differences in $\omega$ between clades are less pronounced (Table 2). Rates of synonymous substitution ($d_s$) appear to be slightly higher in non-passerine birds, both in and outside the PBR, suggesting that it is not solely substitution rates or shorter generation time that is driving the difference in $\omega$ between the groups. We found similar patterns when the data were analyzed using PAML. We found clear evidence for adaptive evolution in songbirds, as evidenced by the lower likelihood score for models that included adaptive evolution compared to those that did not (model 1 vs. 2: $2*$ difference in $\ln L = 291.7$; LRT, $df = 2$, $p < 0.0001$); model 7 vs. 8, $2*$ difference in $\ln L = 299.08$, $df = 2$, $p < 0.0001$). We next examined branch-site and clade models. In model A, the estimated value of $\omega$ for class 2a sites ($0 < \omega_0 < 1$ background vs. $\omega_2 \geq 1$

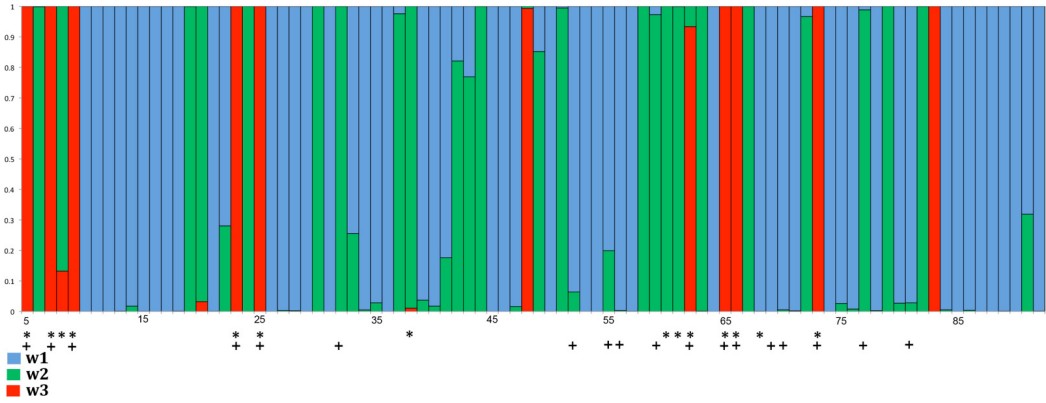

**Figure 4 Distribution of positively selected sites in exon 3 of songbird class I genes as estimated by PAML (model 2).** Red columns indicate the class of sites with a high probability of $\omega > 1$. In this model $\omega_1 = 0.25$ and applies to ~59.3% of the codons (blue); $\omega_2 = 1$ at ~28.2% of the sites (green); and $\omega_3 = 3.53$ at ~12.6% of the sites (red). Asterisks indicate codons assumed to comprise the avian PBR in the MEGA analysis and crosses indicate PBR residues in the human HLA-A2 molecule (*Björkman et al., 1987*; *Saper, Bjorkman & Wiley, 1991*).

foreground) was about 10 times higher in songbirds ($\omega_2$) than in non-passerines ($\omega_0$). For class 2b sites ($\omega_1 = 1$ background vs. $\omega_2 = 1$ foreground), rates of adaptive evolution were about four times higher in songbirds ($\omega_2$) than in non-passerines ($\omega_1$). Model A identified 11 codons with high probabilities of adaptive evolution (see Fig. 4), whether with uniform rates (model 2) or a gamma distribution of rates (model 8) among sites was assumed. This set of codons (numbers 5, 7, 9, 23, 25, 48, 62, 65, 66, 73 and 83) shared nine codons (5, 7, 9, 23, 25, 62, 65, 66 and 73) with those designated in the MEGA analysis. Codons 48 and 83 were subjected to positive selection according to PAML analysis but were not labeled as PBR codons during MEGA analysis. On the other hand, codons 9, 60, 61 and 68, which were labeled as PBR codons in the MEGA analysis, are in close proximity (within one or two codons) to one of the positively selected codons revealed by PAML analyses. In model D, passerines exhibited a level of diversifying selection ($\omega_2$) again about four times higher in passerines than in non-passerines. Model C was the only model that suggested an equivalence of diversifying selection in passerines and non-passerines, with the value of site class 2 (freely varying $\omega_2$) being approximately equal in passerines and non-passerines.

*Shifts in GC content between songbirds and non-passerines*: We noticed conspicuous differences in GC content of our exon 3 sequences between songbirds and non-passerines, particularly for the 3rd codon position (GC3) (Fig. 5). By a standard statistical test, the mean GC3 content differed highly significantly between songbirds and non-passerines (Welch's two sample $t$-test, $t = 10.3108$, $df = 223$, $p$-value $<2.2e-16$; mean passerines 69.77%, mean non-passerines 77.58%), although the GC content overall codon positions did not ($t = -1.4671$, $df = 223$, $p$-value $= 0.1438$; mean passerines 57.23%, mean non-passerines 56.71%). We employed the Brownian motion model of *Thomas, Meiri & Phillimore (2009)* to study the evolution of base composition in our class I sequences while taking phylogeny into account. Surprisingly, using an ultrametricized tree, we found that a model in which GC3 was assumed the same rates of evolution and same mean between

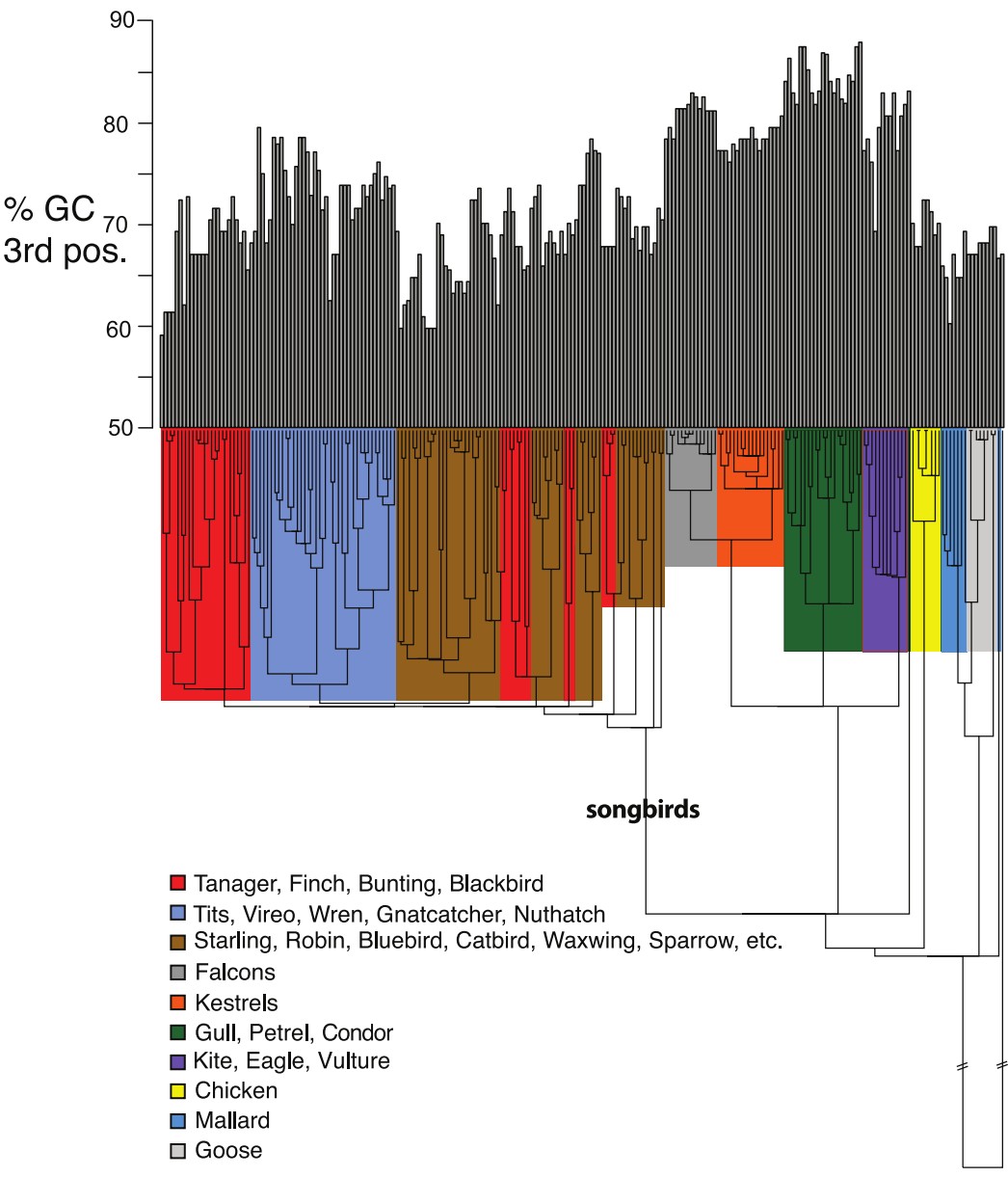

**Figure 5 Base compositional variation in the 3rd codon position of avian class I MHC genes (exon 3).** Clades are indicated according to the key provided. The topology depicted here is a neighbor-joining tree as described in Methods, however, the branch lengths have been ultrametricized as described in Methods to conduct the comparative tests. The branch leading to songbirds is indicated. The topology of this tree differs slightly from that in Fig. 2 because this is a simple neighbor-joining tree, rather than a bootstrap consensus of trees as in Fig. 2.

the two clades was the simplest explanation of the data as assessed by its having the lowest value for the Akaike Information Criterion (AIC; Table 3). Across all the tests, we found no evidence for differences in rates of evolution of GC content in the two clades. However, we reasoned that the long branch leading to the songbird sequences could be causing this lack of significance, because we would expect greater differences between clades and greater

**Table 3  Tests of differences in mean base composition and rates of base compositional change in MHC classs I genes (exon 3) between songbirds and non-passerines using the likelihood model of *Thomas, Meiri & Phillimore (2009)*.** The cells in bold and underlined indicate the model and AIC value that best explains the data under two sets of branch lengths leading to the songbird sequences.

| Model | Length of songbird branch | AIC | Proportion GC – all sites | | | AIC | Proportion GC – 3rd positions | | |
|---|---|---|---|---|---|---|---|---|---|
| | | | Brownian variance | Relative rates (song-birds non-passerines) | Estimated means (song-birds, non-passerines) | | Brownian variance | Relative rates (songbirds, non-passerines) | Estimated means (songbirds, non-passerines) |
| Same mean, same rates | Full | **712.42** | 22.55 | 1, 1 | 56.01 | **993.54** | 78.67 | 1, 1 | 72.28 |
| Different mean, same rates | Full | 714.24 | 22.63 | 1, 1 | 55.79, 56.79 | 993.75 | 78.39 | 1, 1 | 73.55, 67.84 |
| Same mean, Different rates | Full | 714.35 | 22.08 | 1.05, 1 | 56.02 | 995.43 | 80.68 | 0.94, 1 | 72.33 |
| Different mean, different rates | Full | 716.16 | 22.16 | 1.05, 1 | 55.79, 56.79 | 995.65 | 80.27 | 0.94, 1 | 73.55, 67.84 |
| Same mean, same rates | Half | **712.03** | 22.56 | 1, 1 | 56.11 | 993.88 | 78.96 | 1, 1 | 71.69 |
| Different mean, same rates | Half | 713.76 | 22.63 | 1, 1 | 55.79, 56.79 | **993.27** | 78.39 | 1, 1 | 73.55, 67.84 |
| Same mean, different rates | Half | 713.96 | 22.10 | 1.05, 1 | 56.12 | 995.78 | 80.89 | 0.94, 1 | 71.76 |
| Different mean, different rates | Half | 715.69 | 22.17 | 1.05, 1 | 55.76, 56.79 | 995.17 | 80.32 | 0.94, 1 | 73.55, 67.84 |

total Brownian variance with a longer time period separating the two clades in question. We found this to be the case; the significance of a difference in GC3 between songbirds and non-passerines was dependent on the length of the branch leading to the passerine clade. If this branch was cut in half, a model in which the two clades had different means was superior to a model in which they had the same means, not only when rates of evolution were assumed to be the same in the two clades, but even when rates of evolution were assumed to be different, in which case different rates could possibly explain some of the differences in mean (Table 3). Thus the degree to which songbird MHC class I 3rd positions are deemed to differ in their GC content depended on the length of the branch leading to songbirds. This uncertainty has important implications for the analysis of continuous traits in birds.

## DISCUSSION

The complexity of the songbird MHC class I multigene family documented here and in previous studies (e.g., *Westerdahl, 2004*; *Sepil et al., 2012*; *Bollmer et al., 2012*) makes surveying MHC variation through traditional cloning techniques challenging. Along with

its ability to cross-amplify multiple MHC class I loci across a wide diversity of species, an additional advantage of our primer pair is the small size of the PCR amplicons (∼300 bp) that still covers almost the entire coding region of one of the exons comprising the class I PBR – short enough to be amenable to next-generation sequencing but longer than most class I sequences currently available in the databases. These properties make our primers especially suitable for cost-effective pyrosequencing approaches for genotyping large numbers of individuals (see review by *Babik, 2010*).

The high number of divergent sequences isolated from single individuals in our study, despite the low number of clones that we screened per individual, also suggests that our survey was not preferentialy amplifying certain alleles or loci to an extreme degree. Equal amplification of alleles is of great utility when attempting to discriminate between presumably true alleles and those emerging from PCR and/or sequencing artifacts. The exon-intron boundaries where our primers sit are expected to be relatively well conserved within and among related species because their critical role in the process of intron splicing. This expectation would be consistent with the successful cross-amplification and the retrieval of large allele repertoires with our primers. That said, we cannot rule out the existence of polymorphisms in the priming sites that cause poor or non-amplification of certain alleles/loci. Even so, the sequences obtained with our primers are still useful, especially if used in conjunction with previously published primers in other MHC class I regions (e.g., *Westerdahl et al., 2004*; *Sepil et al., 2012*), which together permit evaluation of potential PCR biases in particular taxa.

Our primers successfully amplified a large variety of species of oscine passerines, including representatives from the infraorders Passerida and Corvida (see *Barker, Barrowclough & Groth, 2002*; *Barker et al., 2004* for more details about phylogenetic relationships among passerines). Within Passerida we amplified several members of the core Passeroidea and Muscicapoidea, as well as taxa outside these clades, such as waxwings (Bombycillidae), New world Warblers (Parulidae), nuthatches (Sittidae) and wrens (Troglodytidae). The Red-eyed Vireo (Vireonidae) is nevertheless the unique representative of the clade Corvida in the present study, but because of its basal position within Corvida we predict that these primers should work in other Corvida, such as crows and bowerbirds. Our primers did not amplify, the homologous region from two suboscine passerine species, a manakin (Pipridae) and a New World Flycatcher (Tyrranidae). Thus these primers appear to be songbird-specific, rather than passerine-specific.

This songbird MHC class I data set, when combined with class I sequences from non-passerines, afforded us a macroevolutionary view of MHC class I evolution in birds. We found consistent evidence for stronger adaptive evolution in songbird than in non-passerine class I sequences. The analysis using MEGA, and two of the three PAML models we used gave strong evidence for a higher ratio of nonsynonymous to synonymous ($\omega$) substitutions in the PBR, whether or not PBR codons were chosen a priori. Paradoxically, the only model that did not suggest higher adaptive evolution in the songbird PBR was model C of PAML, which suggested that the intensity of selection in songbirds and

non-passerines was approximately equal. Overall we take these results as strong evidence that songbirds do exhibit a higher rate of adaptive evolution than non-passerines.

This higher intensity of adaptive evolution in songbirds could be part of a general syndrome of songbird evolution driven by their generally small body size, rapid rate of evolution, diversity of habitat occupancy and hence higher encounter rate with parasites. In many studies of molecular evolution, passerines exhibit higher rates of evolution than do non-passerines. This pattern has been attributed to several factors, especially the generation time effect, with passerines having a larger number of generations per unit time than non-passerines. *Nam et al. (2010)* found several predictors of synonymous and nonsynonymous substitution rates in birds. For example, contrary to the predictions of the Hill-Robertson effect, which predicts a positive relationship between the rate of nonsynonymous substitution and chromosome size (due to the increased efficiency of purging presumably deleterious nonsynonymous substitutions on highly recombining small chromosomes), they found a weak negative relationship between $\omega$ and chromosome size in chickens and zebra finches. The MHC class I region is on a microchromosome in chickens, and the single functional class I gene in zebra finches is on the smallest macrochromosome, chromosome 16 (*Ekblom et al., 2011*), which is still larger than a microchromosome. Thus although MHC genes in both chickens and songbirds may be on small chromosomes, their results do not clearly predict which lineage should show higher rates, or indeed may predict that chickens should have higher rates than zebra finches, contrary to our results. MHC class I pseudogenes, of which *Balakrishnan et al. (2010)* found several, are likely dispersed on multiple chromosomes in zebra finch (*Balakrishnan et al., 2010*; *Ekblom et al., 2011*). Thus chromosome position cannot yet inform our understanding of rates of evolution in songbird class I genes.

A more general explanation of higher intensities of selection in songbird MHC class I genes may lie in their larger effective population sizes. The ease with which balancing selection will act on MHC genes, without the negating effect of genetic drift, will depend on the effective population size (*Takahata, 1990*). Larger effective population sizes will allow for positive or balancing selection to act with greater efficiency than smaller effective population sizes; the opposite trend is expected when selection is largely stabilizing or purifying (*Welch, Bininda-Emonds & Bromham, 2008*). Due to their smaller body sizes and shorter generation times, passerines almost certainly have a higher average effective population size than do non-passerines. Ultimately the covariance among all these variables may make it challenging to determine ultimate causes of stronger selection in passerines at MHC class I genes.

The base composition of genes and lineages is an important window in to evolutionary dynamics and can vary among lineages for a variety of genomic and life-history causes (*Nam et al., 2010*; *Nabholz et al., 2011*; *Romiguier et al., 2010*; *Romiguier et al., 2013*; *Backström et al., 2013*). Higher GC content in particular lineages of mammals has been associated with lower body mass and smaller genome size (*Romiguier et al., 2010*). This relationship would predict higher GC content in songbirds than in non-passerines, a prediction recently confirmed by *Backström et al. (2013)* but opposite to what we

found here. Higher GC content in specific genes or chromosomal regions can also be an indicator of increased rates of biased-gene conversion (BGC), a neutral process across much of the genome that can ultimately convert T-A base pairs to C-G base pairs (*Duret & Galtier, 2009*; *Romiguier et al., 2010*). In avian and other MHC genes, gene conversion and recombination have long been known to diversify the PBR regions (*Hess & Edwards, 2002*; *Spurgin et al., 2011*). The weakly higher GC content in the 3rd positions of codons of non-passerine class I genes that we detected here could suggest higher rates of gene conversion in non-passerines. This trend would be somewhat at odds with the higher rates of adaptive evolution observed in songbirds, in so far as gene conversion is thought to be an important source of nonsynonymous mutations in avian MHC genes (*Hosomichi et al., 2008*). In fact, the lower GC content of 3rd positions that we found here contradicts a genome-wide trend for avian genes in which songbirds typically exhibit higher GC content than do non-passerines (although this trend has not been examined in large numbers of species; *Nabholz et al., 2011*; *Backström et al., 2013*). The degree of significance of the higher GC content in 3rd positions of non-passerine class I genes depended on the model used; in standard non-phylogenetic terms, the distinction is highly significant, whereas under a comparative method tailored for continuous phenotypic data, the significance is model-dependent. The study of base compositional variation in avian genes is still in its infancy and the particular base composition of an individual gene may reflect the local rate of recombination in the region of the genome, or it may represent a stochastic effect. Further work in this area is needed.

Our analysis suggested that the significance of the difference in GC3 between songbirds and non-passerines depends on the length of the branch leading to songbirds. This branch length will vary depending on the gene or genomic region analyzed, and will also vary depending on the pattern of evolution displayed by the trait being analyzed, since branch lengths in the phylogeny affect the pattern of covariance among species that is consistent with a Brownian motion model (*Harvey & Pagel, 1991*; *Garland, Bennett & Rezende, 2005*). The analysis of continuous traits such as base composition in birds will depend on a variety of factors influencing the phylogeny and branch lengths of the species being analyzed. As we enter the era of large-scale comparative studies in birds, attention to these diverse factors will enable maximal resolution of evolutionary dynamics.

## CONCLUSION

The main purpose of this study was testing the capability of a newly designed pair of primers to cross-amplify multiple and evolutionarily relevant MHC loci across a broad spectrum of taxa. The order Passeriformes contains thousands of species. Thus, we believe the rapid diffusion of the primers presented here represents a significant advance for the investigation of MHC variation across a widely diverse group of birds. That said, future studies must investigate in more depth patterns of MHC class I variation within each particular taxa, including the estimation of the minimum number of putatively functional genes, pseudogenes and the extent of genetic variation and level of expression within different loci. Our primers appear to target polymorphic and functional MHC class I

genes, a conclusion also supported by differences in the set of alleles isolated in two house finches and two Eastern Bluebirds. Additionally, the primers amplify almost the entire coding sequence of exon 3 (only excluding the four first codons, which seem to be largely conserved among songbirds), a fragment that is somewhat longer than that typically amplified in MHC class I studies in birds. We expect these primers to facilitate further work on MHC class I genes of songbirds, for both ecological and molecular evolutionary studies.

## ACKNOWLEDGEMENTS

We thank Gavin Thomas, Liam Revell, Chris Organ and Andrew Meade for helpful discussion and advice on comparative methods; Niclas Backström for discussion of base composition; the two reviewers for helpful comments; and T. Lenz for discussion of MHC issues.

### Funding

This research was supported by NSF grant IOS-0923088 to Geoff Hill and SVE. During this research, M.A. was funded by a postdoctoral fellowship from the Ministerio de Ciencia e Innovacion (www.micinn.es) of the Spanish government. The funders had no role in study design, data collection and analysis, decision to publish, or preparation of the manuscript.

### Grant Disclosures

The following grant information was disclosed by the authors:
National Science Foundation: IOS-0923088.
Ministerio de Ciencia e Innovacion. Programa de Ayudas Postdoctorales. Expediente -2008-0041.

### Competing Interests

Scott Edwards is an Academic Editor for PeerJ.

### Author Contributions

- Miguel Alcaide conceived and designed the experiments, performed the experiments, analyzed the data, contributed reagents/materials/analysis tools, wrote the paper.
- Mark Liu conceived and designed the experiments, performed the experiments, contributed reagents/materials/analysis tools.
- Scott V. Edwards analyzed the data, contributed reagents/materials/analysis tools, wrote the paper.

### Ethics

The following information was supplied relating to ethical approvals (i.e. approving body and any reference numbers):
Harvard IACUC protocol number AEP 24-06.

## DNA Deposition

The following information was supplied regarding the deposition of DNA sequences:

GenBank accession numbers KC585518-KC585637.

## Data Deposition

The following information was supplied regarding the deposition of related data:

Dryad: http://dx.doi.org/10.5061/dryad.jr583

Treebase: http://purl.org/phylo/treebase/phylows/study/TB2:S14291.

## Supplemental Information

Supplemental information for this article can be found online at http://dx.doi.org/10.7717/peerj.86.

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
