# Peer review of "Major histocompatibility complex class I evolution in songbirds: universal primers, rapid evolution and base compositional shifts in exon 3"

_PeerJ, doi:10.7717/peerj.86_

## Round 0.1 · original submission · Major Revisions

Overall, the two reviewers were positive about your manuscript. However, they have voiced a substantial number of points that need to be addressed before we can publish your manuscript. In particular, please pay close attention to the remarks regarding potential pitfalls of your approach (see e.g. Reviewer 2's concerns on potential biases in primer, limitations due to sampling strategy, etc.).

·

Basic reporting

No comments

Experimental design

No comments

Validity of the findings

No comments

Additional comments

This study provides an important resource for researchers interested in any number of questions concerning the MHC class I of the passerines. In addition, the authors provide a broad comparison of the evolutionary dynamics between Passeriformes and other Orders and report interesting results that contribute to our general understanding of the evolution of MHC in birds. I think the paper could be published as it is; however, the authors and editor may wish to consider the following relatively minor comments:
My comments, in no particular order:
1. Page 7. L 15-16 (and elsewhere). Please tell the reader why this interests you. The introduction mentions “base compositional shifts” as a window on “mutational pressures”. I realize it is obvious to you, but many readers may not be familiar with why this is important, it’s significance etc. You do this in the discussion, but I think a little more information earlier is appropriate.
2. Page 14 L 3-5. This statement appears contradictory. How can a gene, or paralogues of it (or homologues for that matter) be dispersed among micro and macro chromosomes if you only found one? I assume the qualifier “full length” means Balakrishnan et al. found class I fragments elsewhere? Also, (as I am sure you know), Ekblom (2011) provides evidence for a single class I locus on micrchromosome 16 and no evidence for any dispersion of MHC I to other chromosomes. It seems you disagree with their findings/conclusion. I think this is a case where a “but see Ekblom”, at the very least, should be inserted. I might argue that the jury is still out on whether the MHC-B is on a single or plurality of chromosomes in passerines.
3. Perhaps I am being overly fastidious, but there is probably some amount of variation at the primer sites (even intronic) between, and even within species at class I loci. As part of this paper’s purpose is to encourage and facilitate class I studies in other songbird species, it may be useful to add a word of caution about this. Thus, if one were to use your primers in their focal species, it would behoove them to verify that the primer region is conserved in that species, less they grossly underestimate MHC variation. Perhaps this could be incorporated around L 8 of page 16. Related to this, Page 12 L22 onward, “… a large fraction of the totality …” What does this mean? Most? It’s a vague statement.
4. Page 4 L 3. “… polymorphic of the two”---citation?

Reviewer 2 ·

Basic reporting

No comments

Experimental design

No comments

Validity of the findings

No comments

Additional comments

In the present study Alcaide et al developped primers for the amplificatin of an MHC class I exon involved in antigen binding. The primers appear to amplify in 16 tested species. Subsequent phylogenetic analyses suggest that the rates of adaptive evolution are higher in passerines than in non-passerine birds. Finally, passerines appear to have different base composition at third codon positions (GC3) than non-passerines.

Although I think that the present work should be published, I have a number of major concerns:

1. I do think that the reported primers are highly useful and will help researchers greatly with the isolation of MHC class I genes. However, I am less optimistic than the authors that the primers amplify without bias. I think that the presented data are not sufficient for such a conclusion. I strongly believe that, on contrary, the authors have a big responsibility to call for caution with the blind application of ‘universal’ MHC primers. Their work will definitely stimulate more research on MHC class I in passerines, and evolutionary ecologists will apply the primers assuming that they are a perfect tool, unless they are told that they might not be so. While it may be that they are so, there is still a realistic possibility, that they only amplify a subset of alleles or paralogs, and that downstream analyses get heavily flawed if a potential bias is not carefully considered. I thus think that the authors should give a clear hint at the danger of preferential amplification, and discuss routes by which researchers can come from a set of first sequences to an as good as possible genotyping system.

2. The authors do not apply the standard quality checks for PCR artifacts, which are commonly used in MHC research. I indeed understand that with the often highly duplicated MHC of passerines, this is a challenge, and it may be (or not) that for the purposes of the present study less restrictive criteria can be used. But again, authors should be aware of their responsibility. People will copy on their work… I think that it is important to maintain a high quality standard, and that the authors should therefore explain, why their criteria may be sufficient in the present context, but clearly indicate that for other purposes more stringent criteria are clearly required.

3. I am not sure if the species sampling is ideal for a fair comparison of evolutionary rates between passerines and non-passerines (see detailed comments).

4. I am not an expert with the comparative method, but if my limited understanding does not mislead me, the non-significant outcome might be expected (see detailed comments).

Given these comments, I support publication of the present study after considerable revision. First, I would suggest that the authors provide a more balanced discussion of potential of their primers and provide an explanation of the relaxed sequence quality check. Second, the results concerning evolutionary rates should be carefully discussed in the light of potential problems connected with the species sampling, and with respect to the quality of the data (i.e. relaxed criteria for consideration of sequences and shallow per individual coverage).

In the following I provide detailed comments, which the authors may find useful to further improve their manuscript.

*Abstract*
- You may want to mention that exon 3 encodes part of the PBR
- 16 species out of how many tested?
- Concerning the macro-evolutionary analyses comparing songbirds to other birds I wonder in how far the “conservative” approach with only few clones per species sequenced represents a constraint for the analyses?

*Introduction*
- Recent work shows that the classical view on the origin of antigens presented by class I and class II has to be reconsidered, because cross-presentation appears to be frequent.

*Materials and methods*
- Please provide also the alignment algorithm(s) used.
- How are the tested species distributed over the passerine phylogeny? It would be interesting/important to see this plotted on a phylogenetic tree.
- I do not understand which product was sequenced. Did the authors screen the clones using vector primers, and this product was sequenced? Or did they do minipreps and sequence the plastids directly? This is crucial, because in the former case the PCR error rate can be very important (~60 PCR cycles!). See next comment for why this matters.
- If I understand this right, the authors did not follow the standard procedure to control for sequence artifacts. Usually for MHC sequences to be considered true, they have to be obtained by at least two independent PCR reactions. I do see the difficulty with this in passerines, where with a high number of paralogs it is difficult to get a same allele repeatedly. Nevertheless, especially if no minipreps were carried out, it is easily possible to obtain artifactual sequences with 3 bp differences to true alleles. I think that therefore one has to be extremely careful in the interpretation of unconfirmed sequences. Moreover I am worried about people copying this practice. Although it may perhaps be justified for this particular kind of study, and only in passerines (due to the high number of paralogs), it mustn’t be used in standard MHC work. Authors should in my opinion therefore at least very clearly state why they do so here, and that the practice is not generally advisable, e.g. for population-level studies.
- I am not sure if one can justify discarding sequences from putative pseudo-genes when subsequently estimating evolutionary rates?
- I think authors should use a more state-of-the-art method to infer phylogenetic relationships than NJ.
- How were putative PBR codons defined? I think it is difficult to justify this based on previous studies that found signs of positive selection, but it should rather be done based on biochemical studies of the protein.
- Which hypotheses were tested using codeml? It would be good to have these explicitly formulated out.
- Which tree topology was used for the comparative analysis? And for the positive selection analyses?


*Results*
- Lines 12-13, first Results page: I do not agree that the “criteria by which to accept putatively different alleles are fairly restrictive”, for the reasons pointed out above.
- The observed divergence between sequences is overwhelming. How does it compare e.g. to mammalian MHC class I?
- How did the authors assign pseudo-gene status to sequences? How does this account for PCR artifacts?
- Figure 2: Nodes with support <60% should in my opinion be collapsed.
- In my opinion none of the figures shows visibly the clustering by species. Could the network be colored by species? Or could the authors provide a tree which shows this result?
- The presence of recombination could be tested statistically in SplitsTree.
- Please rather state to what kind of omega values site classes correspond than referring to them with their names.
- Authors missed to refer to Figure 4 in the text. In this figure it would be useful to depict the human PBS.
- Which toplogy(ies) did authors use for their ML tests of positive selection? How do different topologies influence the conclusion of higher rates of adaptive evolution in passerines?
- I strongly wonder how the species sampling and the sequencing strategy (and criteria to retain sequences) influence the conclusion of higher rates of adaptive evolution. First, although passerines are a single monophyletic group/order, which is compared to a number of other orders, one should not forget that they do make up the vast majority of bird species, and are themselves not much younger an order than other orders (one might even ponder on what an order is…). I thus wonder if with the present sampling of passerines and non-passerines the conclusion of higher rates of adaptive evolution in passerines can be justified? How large are the overall phylogenetic distances (not MHC, but some more neutrally evolving genes) within passerines versus those in the studied non-passerines? Might clade model C take these into account somehow and might this explain the non-significant result (I am not familiar with the clade models)?
Could the authors discuss how the sequencing strategy, i.e. sequencing a smaller complement of alleles than would be expected per species, might influence estimation of dN/dS?
- Authors refer to figure 3 where they should refer to figure 5. Please also consistently use Fig. or Figure.
- I am not an expert with the comparative method, but if I understand this right, it is not astonishing but rather expected that with the current data no significant result is reached. This is because the trait and phylogeny are indeed highly correlated, no? All passerines, which are all related to eachother, differ consistently in GC3 from non-passerines. Thus the trait indeed DOES have a strong phylogenetic signal, and if we correct for the latter, significance is ‘lost’. To my (limited!) understanding this may indeed be influenced by the length of the branch leading to the passerines. If this grows longer, the lengths of the terminal branches will become more important in the pairwise comparisons, and will overwhelm the signal from between passerines and non-passerines. But please check this with people who are more familiar with this kind of methods.

*Discussion*
- I can’t see how the authors conclude against a bias during PCR. I think that with the high number of paralogs that can be expected (20 for instance in some passerines, thus 40 alleles!), such a bias would not necessarily be visible with the restricted number of clones that were sequenced. I do rather think that the authors should call for caution. This does by no means mean that the primers are not useful, but I think there is a big danger that researchers assume that all there is gets amplified, while adding some further tests might reveal a bias and help avoiding it.
- Were the 16 oscines the only ones which were tested? I.e. was success 100% in oscines? Again, how big a proportion of oscines do the 16 species represent? I think this is important to mention in order to judge how broadly the primers may be expected to amplify.
- Recent work by Ekblom et al. contrary to the cited reference does not find evidence for MHC being situated outside chromosome 16 in zebra finch.
- I would call for caution with interpretation of the findings in the zebra finch MHC. The fact that only one class I gene can be found in the assembly does in my opinion not mean that there is only one. Simply assembly of repetitive regions, like the MHC, is difficult, probably even for genomes based on Sanger sequencing, if coverage is not enormous.
- If larger effective population size drove MHC evolution in passerines, why then do we find more pseudo-genes and more repetitive regions in the passerine versus non-passerine MHC?

---

## Round 0.2 · accepted · Accept

The reviewer has expressed full satisfaction with the revisions, so I am happy to accept the manuscript.

Reviewer 1 ·

Basic reporting

My previous comments have been well worked into the revised version of this manuscript, and I fully support publication.

Experimental design

No comments

Validity of the findings

No comments

Additional comments

No comments